# Client-centered counseling and facilitation in improving modern contraceptive uptake in urban slum of Karachi Pakistan

Zoha Zahid Fazal[1☯], Noor ul Huda Zeeshan[1], Ghazal Moin[2], Alishan Bachlany[2], Yasir Shafiq[3,4☯], Ameer Muhammad[2]*

1 Medical College, The Aga khan University, Karachi, Pakistan, 2 VITAL Pakistan Trust, Karachi, Pakistan, 3 Department of Pediatrics and Child Health, The Aga Khan University, Karachi, Pakistan, 4 Department of Translational Medicine and Center for Research and Training in Disaster Medicine, Humanitarian Aid and Global Health (CRIMEDIM), Università degli Studi del Piemonte Orientale "Amedeo Avogadro, Novara, Italy

☯ These authors contributed equally to this work.
* ameer.muhammad@vitalpakistantrust.org

**Data Availability Statement:** The de-identified dataset is available with all the key variables.

**Funding:** The authors received no specific funding for this work.

## Abstract

### Background

Population growth in Pakistan necessitates the implementation of comprehensive family planning (FP) initiatives. The adoption of modern contraceptives, especially long-acting reversible contraceptives (LARC), and permanent family planning methods in the country is challenging and has yet to reach an optimal level. These challenges are deeply rooted in the lack of informed decision-making, as well as demographic and maternal obstetric history. Interventions tailored according to women's needs can address the challenges faced by FP programs. This paper presents the findings of the implementation of a client-centered counseling and facilitation approach in an urban slum in Karachi, Pakistan. Such an approach has the potential to inform women and help them make better decisions regarding their health.

### Methods

In Rehri Goth, a slum located in Karachi, client-centered counseling along with facilitation at the facility was implemented to encourage the adoption of any modern contraceptive methods, with a specific emphasis on promoting the use of LARCs and permanent methods (where needed) among married women of reproductive age (MWRA). This approach was integrated into the existing Maternal, Neonatal, and Child Health (MNCH) services established in 2014. During the routine delivery of services, data were collected on various aspects including demographic characteristics, obstetric history, motivation to adopt LARCs, and reasons for refusal.

### Results

A total of N = 3079 eligible MWRA received client-centered counseling, and 60.3% accepted modern contraceptive methods after counseling. Furthermore, 32.5% of these MWRA

**Competing interests:** The authors have declared that no competing interests exist.

adopted LARCs or permanent methods. Factors explaining reluctance to adopt any method by MWRA despite specialized counselling were: age >25 years (AOR:1.28, 95% CI:1.08–1.51), no formal education (AOR:1.58, 95% CI:1.36–1.89), having no decision making role at household (AOR:1.60, 95% CI:1.36–1.89), the desire of female or male progeny (AOR:1.86, 95% CI:1.59–2.25) and age of youngest alive ≥3 years (AOR:1.50, 95% CI:1.22–1.84). Factors explaining adoption of short-term methods instead of LARCs or permanent method were: being resident in high under-five mortality clusters (AOR:1.56, 95% CI:1.14–2.14), maternal age > 25 years (AOR:1.88, 95% CI: 1.47–2.40), no decision-making role (AOR:11.19, 95% CI:8.74–14.34), no history of abortions (AOR:2.59, 95% CI:1.79–3.75), no female child (AOR:1.85, 95% CI:1.30–2.65) and ≤ 2 children (AOR:1.74, 95% CI:1.08–2.81).

## Conclusion

Considering the obstacles mothers face when it comes to accessing extended contraception, public health officials can devise effective strategies that empower MWRA to make well-informed and empowered choices regarding their families and reproductive health.

## Introduction

Enhancing the accessibility of family planning (FP) services is vital for the realization of Sustainable Development Goals (SDGs), particularly in the promotion of gender equity and women's empowerment, the improvement of maternal and newborn health, and the advancement of quality education [1]. As of 2017, approximately 214 million women in low-and middle-income countries (LMICs) desired to prevent pregnancy but did not utilize modern contraceptive methods [2, 3]. These figures are highest in Sub-Saharan Africa and Southern Asia, accounting for 39% of all women in developing regions seeking to avoid pregnancy and 57% of those with an unfulfilled demand for modern contraception [3]. To address the issue of unmet needs, it is crucial to enhance both access to and acceptance of modern contraceptive techniques.

Pakistan has a population of approximately 225 million people, making it the fifth most densely populated country in the world. However, projections indicate that this figure could exceed 300 million by 2040 [2, 4]. FP have huge complexities and sensitivities in Pakistan [5]. Although the government and various partners in the FP domain have endeavored to promote FP, challenges remain in increasing the optimal uptake [5, 6]. An insufficient understanding of family planning and the available methods, cultural and religious barriers that discourage contraception and promote large families, limited availability of quality services and FP supplies, especially in low-income areas, and a lack of female empowerment and reproductive health decision-making power are the major factors contributing to the low modern contraceptive prevalence rate (mCPR) [7, 8]. According to recent data, mCPR in Pakistan is only 25%, and unmet needs are among 17% of married women of reproductive age (MWRA) [9]. Consequently, Pakistan faces challenges in reducing population growth, improving maternal health, and achieving sustainable development [10].

Pakistan committed itself to FP commitments under FP2030 [11]. As part of SGDs, there is a huge intention to increase the proportion of FP needs met by modern contraceptives, especially long-acting reversible contraceptives (LARC) or permanent methods in the country

[12]. To achieve this, a targeted approach aimed at reaching the most marginalized and vulnerable women [13]. One strategy for improving modern contraceptive utilization is to provide client-centered counseling and facilitation at facilities, which helps clients find suitable contraceptive methods, continue to use their chosen methods, and return for further assistance if needed [14]. Client-centered counseling and facilitation can also improve a client's self-concept, build trust in services, and support women's reproductive rights [15]. However, little evidence is available on the implementation of such a strategy to improve the adoption of modern contraceptives in Pakistan, where only 19% of women have access to informed choices [9]. Data is even more scarce in deprived areas, such as urban slums.

The VITAL Pakistan Trust, a non-governmental organization, introduced a client-centered counseling and facilitation approach in Rehri Goth, one of the most impoverished urban slums in Karachi. The counseling approach has been integrated into the existing MNCH service delivery package to improve the utilization of modern contraceptives with specific targets to increase the promotion of LARC or guide MWRA regarding permanent methods where it is crucial. The aims of this paper are: (i) to present the adoption rates of overall modern contraceptive methods as well as LARC or permanent methods after the implementation of client-centered counseling and facilitation by FP service providers; and (ii) to delineate the sociodemographic and maternal factors and the reasons that account for the insufficient adoption rates despite the provision of counseling services.

## Methods

This is a prospective observational study and field data collected during the implementation of a "client-centered counseling" approach was utilized in the synthesis of analysis for this paper. This approach was integrated within the Maternal, Neonatal, and Child Health (MNCH) program of VITAL Pakistan which was launched in 2014 at Rehri Goth [15]. The original MNCH program was named as "Caplow project"; executed to reduce under-five mortality in Rehri Goth. The initial program features of the Caplow project included two monthly married women and under-five surveillance, antenatal and skilled birth service provision, nutrition, immunization, and community-based childcare. The rationale behind selecting this specific community was to demonstrate that significant reductions in child mortality are possible, even in challenging peri-urban contexts. This approach ensured a comprehensive strategy to address various factors associated with child and maternal mortality and poor health indicators. In 2015, client-based counseling for family planning, along with facilitation of family planning services with the aim of increasing the uptake of LARC, was incorporated into the program.

### Client-center counseling and facilitation model

We implemented the Family Planning services with close partnership with Greenstar Social Marketing (GSM) and Koohi Goth Women Hospital (KGWH). The key structure was to approach MWRA at the household level through surveillance, build rapport, and identify family planning needs. Based on this, counseling was provided, and facilitation was offered at the facility to build a comfortable and trusted relationship between women, service providers, and project teams. Three different teams, each comprising social scientists and community health workers, were trained in various aspects of counseling. Teams were provided with an in-depth orientation to counseling techniques. They were trained on 'Standardized Training Manual on Family Planning for Community-based Workers' [16], and 'WHO's Medical Eligibility Criteria Tool (MEC) [17]. Based on the training, the teams sought to understand women's needs, preferences, and values, and tailored counseling and contraceptive recommendations to meet

those individual needs. This approach involves active listening, empathy, and respect for the client's decision making. Furthermore, the teams were guided to assess the obstetric and medical histories. Although the focus of our approach was to build trust in FP services and create awareness about LARC, it also involved liaison with key decision makers at households, such as mothers-in-law and where necessary, involvement of husband as well. However, due to cultural barriers and sensitivities, our focus remained on interacting with MWRA and elderly female members in the household during initial phase of implementation. The goal of client-centered counseling is to empower women to make informed decisions about their reproductive health and to support them in achieving their reproductive goals in a manner consistent with their individual preferences, values, and health risks. After receiving counseling, the women were offered the option of obtaining a modern contraceptive method. Subsequently, the teams conducted follow-up visits on a bi-monthly basis after method administration at home by the same team to monitor and receive feedback on the contraceptive method implemented, thus ensuring optimal ongoing care, and establishing trust. For women who underwent LARC or permanent methods, the follow-up duration was an initial one month. However, for those who received short-term methods, the duration and frequency of follow-up were adapted according to the need and continuity of supplies Through constant learning from the field, we recognized that transportation posed a significant barrier to women's access to family planning services. Therefore, we implemented transportation solutions to facilitate and support women in reaching out to FP services at partner facilities. Provision of all the FP services that MWRA required, as well as access to other MNCH services for herself and her under-five children during this period were free of cost. Women were given transportation services accompanied by CHW, as well as the presence of social scientists at the clinic of the service provider to address communication gaps that might exist between MWRA and providers (Fig 1).

## Setting

The population of Rehri Goth was estimated to be approximately 42,980 residents, as per the 2014 census [18]. The community has an annual birth rate of approximately 1200, and nearly 12,000 MWRA reside in this area. The Department of Pediatrics and Child Health within Aga Khan University has established a surveillance site at Rehri Goth to collect data on key MNCH indicators. The under-five mortality rate is approximately 109, whereas the neonatal mortality rate is 59 per thousand live births [18]. The maternal mortality rate in the area is estimated to be approximately 300 per 100,000 live births [19]. The area is divided into 42 small clusters of 250–300 households in each cluster. Furthermore, there are clusters where most of the fishermen reside, with an under-five mortality rate (U5MR) as high as 200–250 per thousand live births. In contrast, in many non-fishermen clusters, U5MR is below the national average of Pakistan, i.e., 79 per 1000 live births. Therefore, for reporting and monitoring the overall progress of the core Caplow project, the area is divided into high mortality clusters, i.e., with U5MR equal to or greater than 79 (national average of Pakistan), and low mortality clusters, i.e., less than 79 per 1000 thousand live births.

## Population, sample size and sampling approach

All MWRA who were permanent residents of Rehri Goth and currently not on any modern contraceptive methods were eligible. Each team was assigned 12–14 clusters based on the population density as well as the most common ethnicity of the population. Using the line listing from the existing surveillance, the team approached MWRA starting from the first cluster assigned to them and gradually moved to the next cluster. As this program is ongoing, we

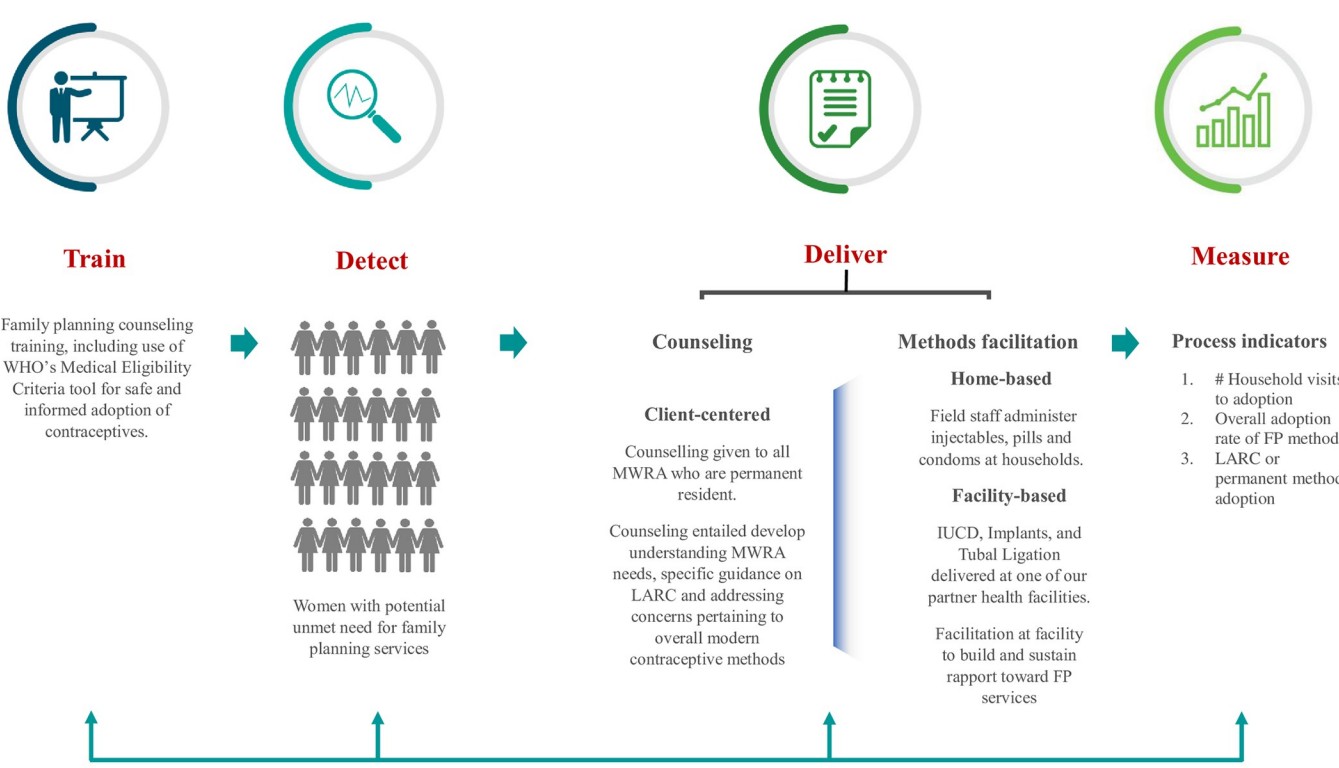

**Fig 1. Implementation model.** The model is explaining key components of the approach.

reported the findings from the initial period of implementation from March 14, 2018, and June 15, 2020.

## Statistical analysis

Frequency and percentage distributions were used to present descriptive analyses of the basic demographic covariates. Adoption rates were calculated based on the total number of MWRA who were approached, expressed as percentages of women who adopted any modern contraceptive methods, and at the second stage, the percentage of those who adopted LARCs or permanent methods. Modern contraceptives include hormone injections (administered every 2 months or 3 months), oral pills, male condoms, intrauterine copper devices (IUCD), post-partum IUCD (PPIUCD), implant (lasts for 3 to 5 years), or permanent method, that is, female sterilization (tubal ligation). Furthermore, in our study, we defined LARC as intrauterine copper devices (IUCD) or postpartum IUCD (PPIUCD), implant (lasts for 3 to 5 years). A Univariate analysis was conducted to assess the determinants of "no adoption of any method at all" and "adoption of only short-term method" (i.e., no LARC or permanent methods) after counseling. Crude odds ratios (CORs) with 95% confidence intervals (CIs) were calculated for statistical analysis. Variables with a p-value less than 0.25 in the univariate analysis were entered into stepwise multivariate logistic regression analysis to calculate adjusted odds ratios (AORs). AORs with a significance level of less than 0.05 are reported. Data analysis was conducted using Stata version 16.

## Ethics approval

The study was approved by the Ethical Review Committee (ERC) (Ref: 001-VPT-IRB-18) of VITAL Pakistan. As the approach was integrated as part of the already proven MNCH intervention, verbal consent was obtained in the local language.

# Results

## Descriptives and baseline characteristics of MWRA

The study approached a total of 3,736 women of reproductive age, which represents approximately 31.1% of the total population of women of reproductive age in Rehri Goth, during the reporting period. Out of these participants, 82.2% (n = 3,079) were eligible for counseling, while the remaining 17.8% (n = 667) either refused counseling at the initial stage or were already using modern contraceptives. Fig 2, impact of client-centered counseling.

Women who refused counseling cited lack of family support and religious beliefs as the main reasons. The average age of the participants was 29.80 ± 7.73 years, and 46.5% (n = 1,432) belonged to high mortality clusters. About two-thirds of the eligible women (65.8%, n = 2,027) had more than two live births, and 62.0% (n = 1,911) had no formal education. (Table 1).

## Modern contraceptives uptake

The overall acceptance rate of any modern contraceptive method after counseling was found to be 60.3% (n = 1,856). LARC or permanent method adoption rate accounted for more than

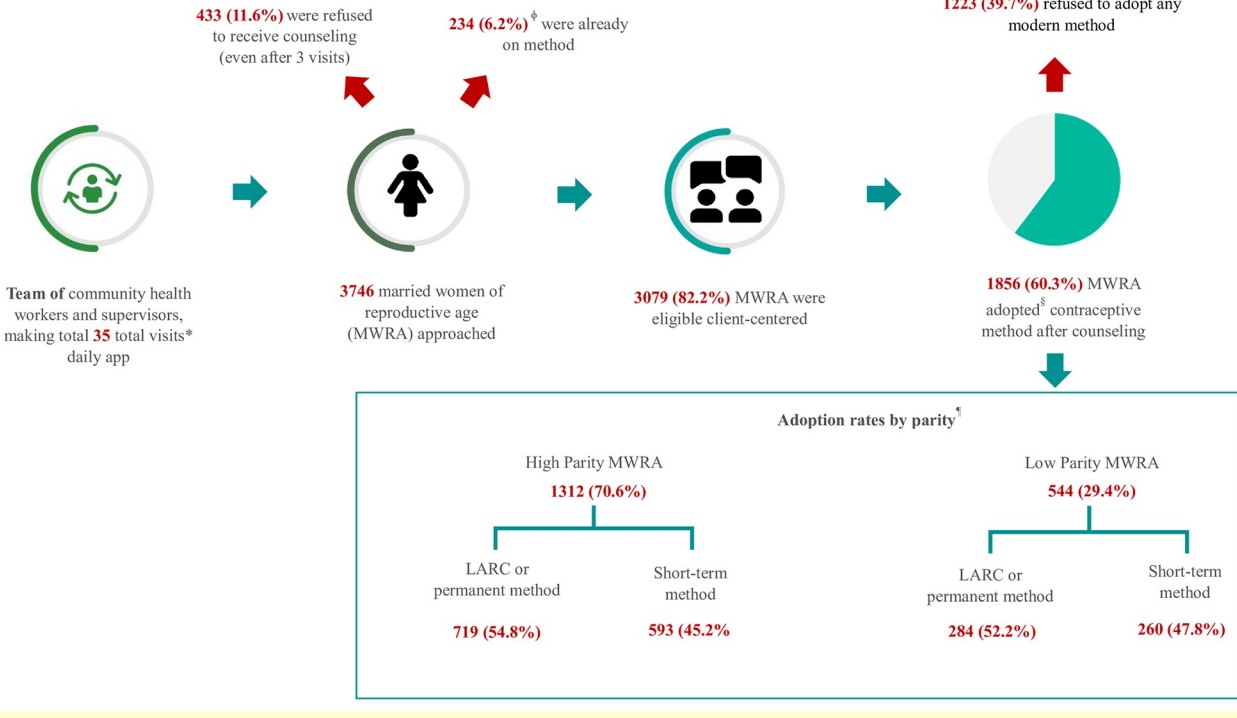

♦ Out of 234, 90.3% were already using short-term and 9.7% LARC or permanent methods
§ Definition of adoption is drawn from the FP 2020 literature; it refers to women using contraception for the first time in their lives and resuming use after a hiatus.
¶ High parity refers to women who have 2 or more previous live births at the time they were approached
* Visit to convince to receive counseling, follow-ups in case of refusal or meet with spouse or extended, pick, and drop to and from FP services and follow-up post administration of method

**Fig 2. Impact of client-centered counseling.** The flow of participants in a study.

**Table 1. Baseline characteristic of MWRA.**

| Characteristics | Value |
|---|---|
| **Demographic information** | |
| Maternal age–Mean ± SD | 29.80 ±7.73) |
| Maternal education–N (%) | |
| *No formal education* | 1911 (62.1) |
| *Primary* | 891 (28.9) |
| *Secondary or above* | 277 (9.0) |
| **Obstetric history—Mean ± SD** | |
| *Gravida* | *4.07 ±2.48* |
| *Parity* | *3.89 ±2.36* |
| *No. of children currently alive* | *3.43 ±2.06* |
| *Age of eldest child currently alive (years)* | *9.09 ±5.90* |
| *Age of youngest child currently alive (years)* | *1.50 ±2.54* |
| **Overall adoption rate to contraceptive–N (%)** | |
| *Any modern contraceptive methods adopted* | *1856 (60.3)* |
| *Overall LARCs or permanent method\* adopted* | *1003 (32.5)* |
| **Distribution of type of FP methods adopted–N (%)** | |
| *Implants* | *579 (31.2)* |
| *IUCDs* | *253 (13.6)* |
| *PPIUCDs* | *37 (2.0)* |
| *Tubal ligations* | *134 (7.2)* |
| *Injectables* | *563 (30.3)* |
| *Oral tablets* | *215 (11.6)* |
| *Condoms* | *75 (4.0)* |

\* N = 1003 adopted LARCs or permanent method. This translated into 54.0% LARCs or permanent method adoption out of those who applied any modern methods and 32.5% of total eligible women who received counseling

half of those who adopted any modern contraceptive method after counseling, and it was 32.5% (n = 1,003) of the total eligible MWRA. Implants were the most applied LARC method, representing 31.2% (n = 579), while the most accepted method was hormone injection, accounting for 30.3% (n = 563) of the total modern contraceptive method uptake (Table 1). High-parity women (those who had at least 2 alive births in the past) had a slightly higher acceptance of LARC or permanent methods, with 54.8% compared to 52.2% among low-parity women (Fig 2).

## Factors associated with no uptake of any modern contraceptives

The study found that several factors were associated with poor uptake of contraceptive methods despite counseling. Women aged over 25 years had a higher chance of poor uptake (45.5%) compared to those under 25 years (34.3%) (AOR: 1.28, 95% CI: 1.08, 1.51). Women with no formal education also had a lower uptake (44.3%) compared to those with some formal education (32.3%) (AOR: 1.58, 95% CI: 1.36, 1.89). Additionally, uptake was lower among women who were not involved in household decision-making (43.9%) compared to those who were involved in decision-making (31.6%) (AOR: 1.60, 95% CI: 1.36, 1.89). Interestingly, women who had no female child had poorer uptake (52.1%) compared to those who had at least one female child alive (36.4%) (AOR: 1.86, 95% CI: 1.59, 2.25), and women with no male child had 55.2% poor uptake compared to 38.3% who had at least one male child (AOR: 2.05,

95% CI: 1.53, 1.44). Finally, women with the youngest child alive aged three years or more had 45.3% poor uptake compared to 38.5% in the other group (AOR: 1.50, 95% CI: 1.22, 1.84).

In summary, this study highlights the need for targeted interventions to improve contraceptive uptake among women who are older, have no formal education, are not involved in household decision-making, have no female or male child, or have the youngest child aged three years or more. Such interventions should be tailored to address the specific needs of these women and to ensure their access to high-quality family planning services. (Table 2A).

## Factors associated with the uptake of short-term methods only

The factors associated with the adoption of short-term methods instead of LARC or permanent methods include the high under-five mortality clusters where women reside, maternal age, involvement in household decision making, history of miscarriages, gender of the currently alive child, and the total number of children alive. MWRA residing in high under-five mortality clusters were more likely to adopt short-term methods (50.8%) compared to those in low mortality clusters (42.2%) (AOR: 1.56, 95%CI: 1.14, 2.14). Women above the age of 25 were more likely to adopt short-term methods (52.1%) than women aged 25 or younger (41.2%) (AOR: 1.88, 95%CI: 1.47, 2.40). Women who were involved in household decision making were more likely to adopt short-term methods (64.7%) compared to those who were not involved (15.9%) (AOR: 11.19, 95%CI: 8.74, 14.34). MWRA with no history of miscarriage were more likely to adopt short-term methods (47.2%) compared to those with a history of miscarriage (35.1%) (AOR: 2.59, 95%CI: 1.79, 3.75). Women with no female child were more likely to adopt short-term methods (54.2%) compared to those with at least one female child (44.3%) (AOR: 1.85, 95%CI: 1.30, 2.65). Lastly, MWRA with 2 or fewer children alive were more likely to adopt short-term methods (49.4%) compared to those with more children alive (44.1%) (AOR: 1.74, 95%CI: 1.08, 2.81) (Table 2B).

## Motivation of adoption of any method

Birth spacing was the most common reason for adopting modern contraceptive methods after receiving counseling, with N = 1468 women (79%) reporting this as their motivation. The second most common reason was existing medical conditions, which prompted N = 212 women (11%) to make a more informed decision about adopting contraceptive methods. Finally, N = 176 women (10%) reported being motivated to adopt modern contraceptive methods for birth control (Fig 3A).

## Reasons for refusals

There are various reasons for rejection of contraceptive methods. The most prevalent factor is familial restrictions, as reported by N = 355 women (29%). Following this is the desire to have male children within the household, as indicated by N = 315 women (26%). The third most common reason is religious beliefs, which hindered uptake, as documented by N = 250 women (20%). Additionally, documented reasons included the desire for more children by N = 183 women (15%), fear of the side effects of modern contraceptive methods by N = 107 women (9%), and a reported desire for female children by N = 13 women (1%) (Fig 3B).

## Discussion

Our preliminary study results demonstrated that client-based counseling, embedded with the facilitation of service uptake at the facility and post-method administration follow-up, has yielded positive outcomes and impacts the overall uptake of modern contraceptives. The

**Table 2. Factors associated with no uptake at all or uptake only short-term FP methods.**

| | 2a\| Factors associated with no uptake of any modern contraceptive methods | | | | | | 2b \| Factors associated with uptake of only short-term methods | | | | | | |
|---|---|---|---|---|---|---|---|---|---|---|---|---|---|
| | MWRAs received counseling N (%) | % MWRA who did not adopted FP | Crude odd ratio (95% CI) | p-value | Adjusted odd ratio (95%CI) | p-value | MWRAs adopted any method N (%) | % LARCs or permanent method adopted | % Only short-term method adopted | Crude odd ratio (95% CI) | p-value | Adjusted odd ratio (95%CI) | p-value |
| **Cluster Type** | | | | | | | | | | | | | |
| Low under-five mortality clusters | 1,647 (53.5) | 36.8 | Ref | | | | 1,041 (63.2) | 57.8 | 42.2 | Ref | | | |
| High under-five mortality clusters | 1,432 (46.5) | 43.1 | 1.30 (1.12, 1.50) | <0.00 | | | 815 (56.9) | 49.2 | 50.8 | 1.41 (1.17, 1.70) | <0.00 | 1.56 (1.14, 2.14) | 0.005 |
| **Language** | | | | | | | | | | | | | |
| Other languages | 1,154 (37.5) | 35.8 | Ref | | | | 741 (64.2) | 56.5 | 43.5 | Ref | | | |
| Sindhi | 1,338 (43.5) | 42.9 | 1.34 (1.15, 1.58) | <0.00 | | | 764 (57.1) | 49.5 | 50.5 | 1.32 (1.08, 1.62) | 0.006 | | |
| Pashtun | 587 (19.1) | 40.2 | 1.20 (0.98, 1.48) | 0.072 | | | 351 (59.8) | 58.7 | 41.3 | 0.91 (0.70, 1.18) | 0.504 | | |
| **Age of Woman** | | | | | | | | | | | | | |
| <25 year | 1,590 (51.6) | 34.3 | Ref | | | | 1,044 (65.7) | 58.8 | 41.2 | Ref | | | |
| 25 years and above | 1,489 (48.4) | 45.5 | 1.59 (1.38, 1.84) | <0.00 | 1.28 (1.08, 1.51) | 0.003 | 812 (54.5) | 47.9 | 52.1 | 1.55 (1.29, 1.86) | <0.00 | 1.88 (1.47, 2.40) | <0.00 |
| **Maternal education** | | | | | | | | | | | | | |
| Some formal education | 1,168 (37.9) | 32.3 | Ref | | | | 791 (67.7) | 55.8 | 44.2 | Ref | | | |
| No formal education | 1,911 (62.1) | 44.3 | 1.67 (1.43, 1.94) | <0.00 | 1.58 (1.34, 1.86) | <0.00 | 1,065 (56.1) | 52.8 | 47.2 | 1.12 (0.93, 1.35) | 0.202 | | |
| **Women part of decision making** | | | | | | | | | | | | | |
| Woman is a part of decision making | 1,041 (33.8) | 31.6 | Ref | | | | 712 (68.4) | 84.1 | 15.9 | Ref | | | |
| Woman is not a part of decision making | 2,038 (66.2) | 43.9 | 1.69 (1.45, 1.98) | <0.00 | 1.60 (1.36, 1.89) | <0.00 | 1,144 (56.1) | 35.3 | 64.7 | 9.70 (7.67, 12.27) | <0.00 | 11.19 (8.74, 14.34) | <0.00 |
| **Gravidity** | | | | | | | | | | | | | |
| > 2 | 2,069 (67.2) | 35.7 | Ref | | | | 1,331 (64.3) | 54.9 | 45.1 | Ref | | | |
| 2 or less | 1,010 (32.8) | 48.0 | 1.67 (1.43, 1.94) | <0.00 | | | 525 (52.0) | 51.8 | 48.2 | 1.13 (0.92, 1.38) | 0.226 | | |
| **Parity** | | | | | | | | | | | | | |
| > 2 | 2,027 (65.8) | 35.3 | Ref | | | | 1,312 (64.7) | 54.8 | 45.2 | Ref | | | |

*(Continued)*

**Table 2.** (Continued)

| | n (%) | % | OR (CI) | p | aOR (CI) | p | n (%) | % | % | OR (CI) | p | aOR (CI) | p |
|---|---|---|---|---|---|---|---|---|---|---|---|---|---|
| 2 or less | 1,052 (34.2) | 48.3 | 1.71 (1.47, 1.99) | <0.00 | | | 544 (51.7) | 52.2 | 47.8 | 1.11 (0.90 1.35) | 0.307 | | |
| **Stillbirths in the past** | | | | | | | | | | | | | |
| At least 1 | 916 (29.7) | 37.3 | Ref | | | | 574 (62.7) | 53.3 | 46.7 | 1.04 (0.85, 1.27) | 0.672 | | |
| None | 2,163 (70.3) | 40.7 | 1.15 (0.98, 1.35) | 0.079 | | | 1,282 (59.3) | 54.4 | 45.6 | Ref | | | |
| **Miscarriages in the past** | | | | | | | | | | | | | |
| None | 2,746 (89.2) | 39.4 | Ref | | | | 1,665 (60.6) | 52.8 | 47.2 | 1.65 (1.21, 2.26) | 0.002 | 2.59 (1.79, 3.75) | <0.00 |
| At least 1 | 333 (10.8) | 42.6 | 1.14 (0.91, 1.44) | 0.249 | | | 191 (57.4) | 64.9 | 35.1 | Ref | | | |
| **Female child alive** | | | | | | | | | | | | | |
| At least 1 | 2,432 (79.0) | 36.4 | Ref | | | | 1,546 (63.6) | 55.7 | 44.3 | Ref | | | |
| None | 647 (21.0) | 52.1 | 1.89 (1.59, 2.25) | <0.00 | 1.86 (1.48, 2.34) | <0.00 | 310 (47.9) | 45.8 | 54.2 | 1.48 (1.16–1.89) | 0.001 | 1.85 (1.30, 2.65) | 0.001 |
| **Male child alive** | | | | | | | | | | | | | |
| At least 1 | 2,818 (91.5) | 38.3 | Ref | | | | 1,739 (61.7) | 54.5 | 45.5 | Ref | | | |
| None | 261 (8.5) | 55.2 | 1.98 (1.53, 2.56) | <0.00 | 2.05 (1.53, 1.44) | <0.00 | 117 (44.8) | 47.9 | 52.1 | 1.30 (0.89, 1.89) | 0.167 | | |
| **Total number of children alive** | | | | | | | | | | | | | |
| > 2 | 1,841 (59.8) | 34.8 | Ref | | | | 1,200 (65.2) | 55.9 | 44.1 | Ref | | | |
| 2 or less | 1,238 (40.2) | 47.0 | 1.66 (1.43, 19.2) | <0.00 | | | 656 (53.0) | 50.6 | 49.4 | 1.23 (1.02, 1.49) | 0.028 | 1.74 (1.08, 2.81) | 0.022 |
| **Age of eldest child** | | | | | | | | | | | | | |
| Less than 5 years | 1,923 (62.5) | 36.3 | Ref | | | | 1,224 (63.7) | 54.0 | 46.0 | 1.00 (0.82, 1.21) | | | |
| 5 years or above | 1,156 (37.5) | 45.3 | 1.45 (1.25, 1.68) | <0.00 | | | 632 (54.7) | 54.1 | 45.9 | Ref | 0.964 | | |
| **Age of youngest child** | | | | | | | | | | | | | |
| Less than 3 years | 2,520 (81.8) | 38.5 | Ref | | | | 1,550 (61.5) | 55.5 | 44.5 | Ref | | | |
| 3 years or above | 559 (18.2) | 45.3 | 1.32 (1.09, 1.58) | 0.003 | 1.50 (1.22, 1.84) | <0.00 | 306 (54.7) | 46.4 | 53.6 | 1.44 (1.12, 1.84) | 0.003 | | |

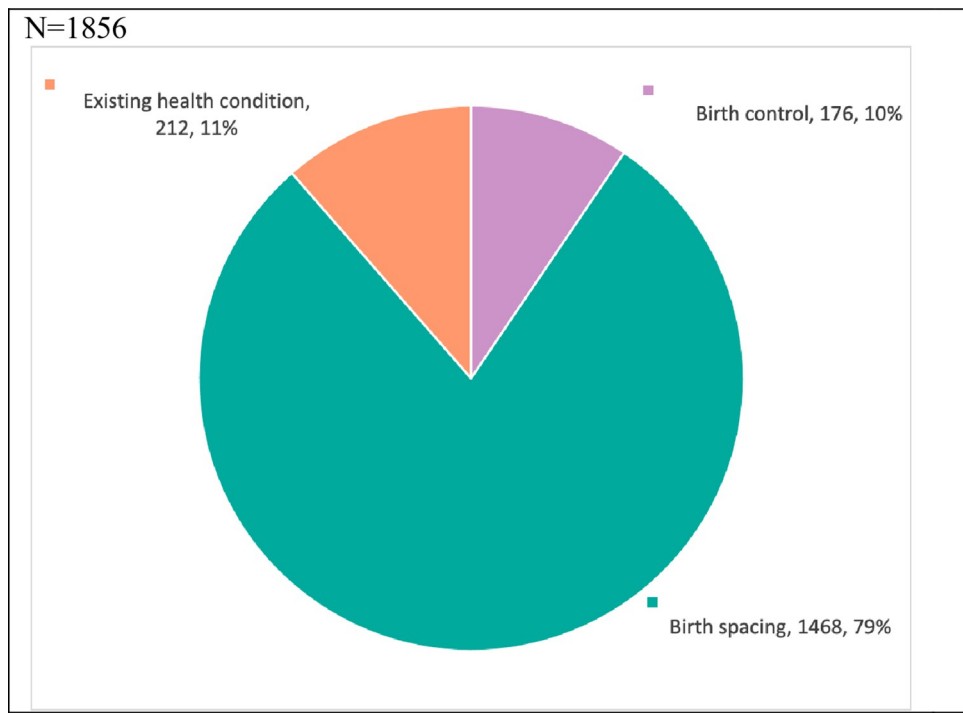

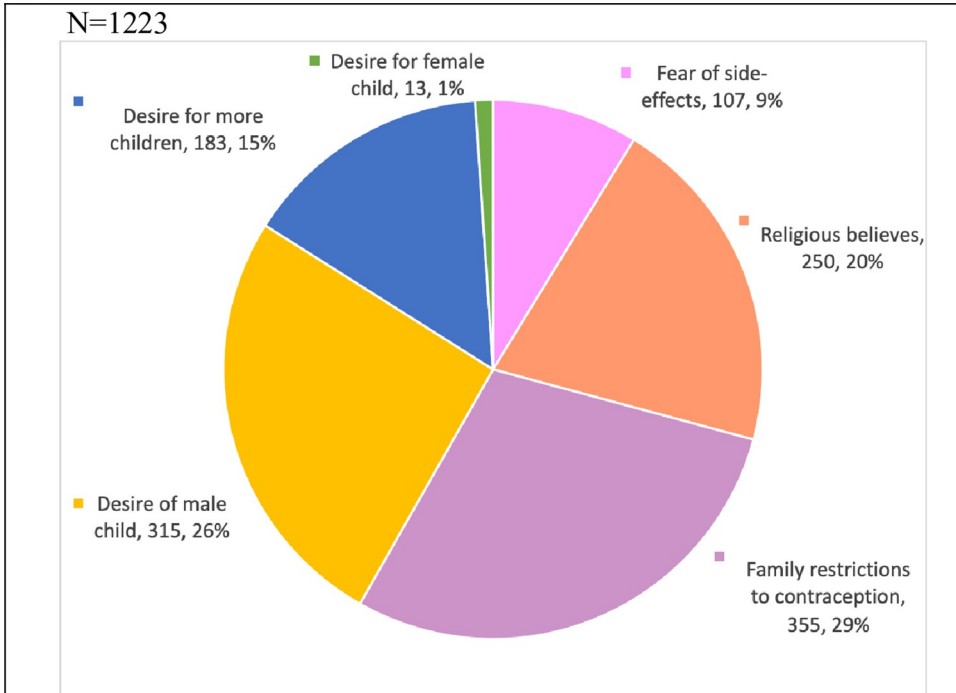

**Fig 3.** **a.** Motivation to adopt any modern contraceptive method. Presenting data of MWRA who accepted the modern methods, **b.** Reasons to refuse any modern contraceptive method. Presenting data of those who did not accept any method after counseling.

existing literature supports such counseling approaches and has shown that they are linked to higher acceptance rates of modern methods [20]. Interestingly, our data suggest that only 6.2% of the MWRA used modern contraceptives when our teams approached them, and only 9.4% used LARC or permanent methods.

A previous study in Karachi with a similar approach, using door-to-door services by community health workers to provide quality family planning services at service providers, resulted in a 42.6% adoption rate for modern contraceptives at the endline, compared to 30.1% at baseline [21]. Although we did not have a baseline comparison, our overall uptake of 60.2% was a great success for the model implemented at Rehri Goth. Furthermore, the adoption of LARC or permanent methods was more than double in our study, at 32.5%, compared with a previously reported study showing only 14.9% acceptance rates [21]. In addition, another study from the neighboring communities of Karachi, Pakistan, showed only a 2.4% increase in mCPR among those who received home-based family planning counseling and referral for modern contraceptive use compared to those who did not [22]. In Pakistan, the mCPR remains low, at less than 3% for implants and IUCD [9]. Client-centered counseling has proven to be a successful approach if implemented with the intention of understanding community needs and involves a continuous learning process to adopt field lessons [23–25]. Our model and implementation strategy stand uniquely for the setting. Several features of the model make this successful. Firstly, integration of client-centered counseling within the existing comprehensive MNCH program helped in building trust of MWRA on the FP. Services. The success of our model lies in its unique integration with MNCH programs through well-trained social scientists and CHWs, coupled with the facilitation of women at service providers by teams, which has helped build trust at the community level and has generated a demand for implants, IUCD and PPIUCD. Such distinctiveness was missing in the previously reported studies, where focus was not LARC but more general, integration within MNCH program was not the scope and facilitation through well-trained social scientists was missing [21, 22].

Maternal education, age, and contextual risks play a pivotal role in the factors that may be associated with low uptake of modern contraceptives. Information about modern contraceptive methods, including their effectiveness, safety, and side effects, may not be readily available to older women [26, 27]; poor literacy also complements this. Some cultures consider discussing contraception taboo or inappropriate, making it difficult for women to obtain information and advice regarding contraception. Older women had less knowledge of contraceptive methods than younger women, with most of their knowledge being acquired through casual sources such as friends and relatives [28]. Consequently, this generates many myths and misconceptions pertaining to modern contraceptive methods, especially LARC [29]. Contextually, MWRA from high-mortality clusters of Rehri, where more than half of the population is a fishermen community [18], has the worst health indicators and poor health care seeking. In high mortality clusters of Rehri Goth, the literacy rate is low, that is, 39.4% compared to 40.0% in low mortality clusters; women are less likely to be involved in decision making (33.1% compared to 34.4%), and have more children (i.e., parity greater than 2 is 35.1% compared to 33.5% in low morality clusters). Therefore, socio-cultural, and individual factors may impose unique challenges in the adoption of modern methods, as reported by other researchers [28–32].

There is a significant impact on women's access to and use of modern contraceptives when they lack decision-making power regarding their own reproductive health [30, 31]. In situations where women are unable to make informed decisions about their health, they may not be able to access contraception. They may be forced to use a method that does not meet their needs [32, 33]. When it comes to making decisions about contraception, women are often expected to defer their husbands or other male family members and gender power dynamics that prevent them from making decisions about their reproductive health [34]. Furthermore,

cultural and social norms often stigmatize women who use contraception, which can discourage them from seeking and using these methods [34, 35]. This is particularly true for the women in Rehri Goth. These factors resulted in a lack of choices toward modern contraceptive methods, predominantly LARC, and poor decision-making for birth spacing without considering the health consequences of having more children.

Considering women's obstetric characteristics, those with a history of abortion may be reluctant to use modern contraceptives because of negative experiences or concerns regarding side effects. It is noteworthy that some women who have had abortions may believe that they have a reduced probability of conceiving again, causing them to choose not to use contraception. Moreover, the age of the youngest child was also considered when adopting modern contraceptives. The use of contraceptives may be delayed until women are ready to have another child if they have young children. The desire for more children can also impact modern contraceptive uptake [33–35]. Women who desire more children may be less likely to use modern contraceptives or use them inconsistently, as they perceive contraceptive use as a barrier to achieving their fertility goals [34]. Similarly, women who do not desire more children may feel that contraception is unnecessary [34–36].

The gender of currently living children is also an influential factor [37, 38]. In many contexts, there is a preference for having male children for various reasons, such as inheritance practices, continuation of the family name, or cultural norms [37–39]. This preference may lead to a higher desire for contraception after the birth of a son, as families may want to limit or limit the number of future pregnancies in the hope of having more sons [40]. Our findings are also the same, in that women with no male children are more prone to reject the concept of modern contraception. Surprisingly, we also found that women with no female children are less likely to adopt modern contraceptive methods, including LARC. These findings may contradict the existing literature From Pakistan, but data from other countries support this fact [41, 42]. The literature suggests that families with no daughters may prefer to have more children and delay contraception [41, 42], especially among households in which women have formal education [42].

By engaging with women in need continuously, our approach appeared to be effective in increasing confidence in counseling and recommending modern contraceptives, particularly LARCs, as the preferred method. Additionally, periodic follow-ups after the adoption of any contraceptive method have further strengthened provider-client relationships and increased levels of trust. A key element of our model is active listening and the involvement of MWRA in the decision-making process. Considering that male participation may lead to more significant results, this feature will be incorporated into the future scale of the program. While some of the findings in this study may overlap with the existing literature on the client-centered approach, our contribution lies in proposing strategies tailored to a specific community. One important aspect of our findings was the difference in the adoption rate between high- and low-mortality clusters, which may be crucial to understanding and showing that disparities exist even within urban slums. Therefore, identifying context-specific interventions along with in-depth knowledge of the community that can promote women's empowerment while navigating the challenges posed by a lack of awareness, information about LARC, and socio-cultural constraints within the same geography is important. Understanding these challenges is crucial. By addressing these gaps in the literature, our study provides valuable insights for practitioners and researchers working in similar contexts [43]. An important step forward is digital and data integration to identify obstetric and health risks using real-time information from the surveillance to help teams to provide more informed support to the women, which VITAL has recently implemented in the MNCH and FP program.

Despite the success of this approach, it has several limitations. First, although the program team can encourage women to adopt modern contraceptive methods, the final decision ultimately rests with the provider of the facility. Contraceptive providers may hold biases that can affect their ability to provide comprehensive and unbiased counseling for women seeking modern contraceptives. These biases may stem from the conventional way of training that they might have received in the past, personal values, or limited exposure to diverse populations from areas such as Rehri Goth. Inadequate, outdated, or poorly designed training programs can also hinder providers from delivering accurate, comprehensive, and client-centered counseling and services [44–46]. To address these limitations, a standardized training mechanism was developed, and a continuous feedback mechanism was introduced. This helped to create synergy with the program and across partners. Furthermore, there was a shortage of supplies for IUCDs and implants during the implementation period, which may have resulted in accessibility issues for a limited period. Additionally, the program structure failed to sufficiently address the importance of male involvement and communication between spouses during contraceptive counseling and acceptance which will be more inclusive in the next phase of this model. Finally, we collected data on a very limited set of variables that we thought were important for the context and convenient to collect.

## Conclusion

The integration of a client-centered approach with comprehensive MNCH services, facilitation/follow-ups, and the provision of free and continuous supplies has significant benefits for women in vulnerable communities. Adapting methodologies to align with unique cultural dynamics is crucial, and context-specific adaptations can enhance women's empowerment and address their power dynamics. Data integration is also important for frontline healthcare workers to access timely information. Therefore, future research should focus on evaluating the impact of integrated MNCH services and the use of real-time data to identify risks and needs within a client-centered framework that can help health workers support women in informed decision-making in accessing modern contraceptives.

## Acknowledgments

The authors would like to thank the community, GSM and KGWH for their continuous involvement and support, and the teams for their contributions.

## Author Contributions

**Conceptualization:** Yasir Shafiq, Ameer Muhammad.

**Data curation:** Alishan Bachlany.

**Formal analysis:** Ghazal Moin, Yasir Shafiq, Ameer Muhammad.

**Software:** Alishan Bachlany.

**Supervision:** Ameer Muhammad.

**Writing – original draft:** Zoha Zahid Fazal, Yasir Shafiq, Ameer Muhammad.

**Writing – review & editing:** Noor ul Huda Zeeshan, Yasir Shafiq, Ameer Muhammad.

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
