## [Decision Letter · Decision Letter 0]

23 Jan 2023

PONE-D-22-17428Reproductive Health for Women in Need – A Novel Perspective of a Case from Urban Slum of Karachi Using Client-centered Counseling and Facilitation of High-risk Women in Improving Modern Contraceptive Prevalence RatePLOS ONE

Dear Dr. Ameer Muhammad,

Thank you for submitting your manuscript to PLOS ONE. After careful consideration, we feel that it has merit but does not fully meet PLOS ONE’s publication criteria as it currently stands. Therefore, we invite you to submit a revised version of the manuscript that addresses the points raised during the review process.

ACADEMIC EDITOR    The results does not answer the objectives. what are the results of client centered  facilitation ?  please explain the process of counselling in detail.  multiple Grammatical mistakes. 

We look forward to receiving your revised manuscript.

Kind regards,

Sidrah Nausheen, FCPS

Academic Editor

PLOS ONE

Journal Requirements:

 “No funding”

“No author have competing interests”

4. Please amend the manuscript submission data (via Edit Submission) to include authors Rashid Ali and Alishan Bachlany.

6. Please include your tables as part of your main manuscript and remove the individual files. Please note that supplementary tables (should remain/ be uploaded) as separate "supporting information" files"

Reviewers' comments:

Reviewer's Responses to Questions

**Comments to the Author**

1. Is the manuscript technically sound, and do the data support the conclusions?

Reviewer #1: Yes

Reviewer #2: No

2. Has the statistical analysis been performed appropriately and rigorously? 

Reviewer #1: Yes

Reviewer #2: I Don't Know

3. Have the authors made all data underlying the findings in their manuscript fully available?

Reviewer #1: Yes

Reviewer #2: Yes

4. Is the manuscript presented in an intelligible fashion and written in standard English?

Reviewer #1: Yes

Reviewer #2: No

5. Review Comments to the Author

Reviewer #1: The study is an interesting read and the draft is well written. However, there are few suggestions for the improvemrnt:

1. Figure 1 is not clearly visible and should be improved.

2. The maternal education is a binary variable with two categories "no education" and "atleast primary". These categories are not exhaustive and do not include mothers with 1-5 education levels. This variable needs to be re- constructed.

3. Conclusion section needs to be improved. The objectives of the study should be matched with its findings and mention limitations, if any.

Reviewer #2: Authors have chosen a very pertinent topic especially in the context of a LMIC like Pakistan.

However, Objectives of the study are vague and not stated clearly.

Methodology is not clear

Results of the study do not match the mentioned objectives

6. PLOS authors have the option to publish the peer review history of their article (what does this mean?). If published, this will include your full peer review and any attached files.

Reviewer #1: No

Reviewer #2: No

---

## [Author Response · Author response to Decision Letter 0]

24 Feb 2023

Respected editor and reviewers,

Greetings.

Many thanks for your feedback. We have thouroughly looked into the comments and addressed all of them. Our point-by-point response is attached in a file. 

We are looking forward in case of further comments

Many thanks

Regards

Ameer Muhammad

---

## [Editor Report · Decision Letter 1]

8 Mar 2023

PONE-D-22-17428R1Reproductive Health for Women in Need – A Novel Perspective of a Case from Urban Slum of Karachi Using Client-centered Counseling and Facilitation of High-risk Women in Improving Modern Contraceptive Prevalence RatePLOS ONE

Dear  Ameer Mohammad <table border="0" cellpadding="0" cellspacing="0" class="datatable3" style="border-collapse: collapse; width: 678px; line-height: 14px; color: rgb(0, 0, 51); font-family: verdana, geneva, arial, helvetica, sans-serif; font-size: 11.2px;"> 

</table>,

Thank you for submitting your manuscript to PLOS ONE. After careful consideration, we feel that it has merit but does not fully meet PLOS ONE’s publication criteria as it currently stands. Therefore, we invite you to submit a revised version of the manuscript that addresses the points raised during the review process.

    we strongly propose a major revision of this article as there are major flaws. the objectives are not addressed in results and are not clear. Please submit your revised manuscript by 15th April 23 . If you will need more time than this to complete your revisions, please reply to this message or contact the journal office at plosone@plos.org. Please include the following items when submitting your revised manuscript:A rebuttal letter that responds to each point raised by the academic editor and reviewer(s). You should upload this letter as a separate file labeled 'Response to Reviewers'.A marked-up copy of your manuscript that highlights changes made to the original version. You should upload this as a separate file labeled 'Revised Manuscript with Track Changes'.An unmarked version of your revised paper without tracked changes. You should upload this as a separate file labeled 'Manuscript'.

We look forward to receiving your revised manuscript.

Kind regards,

Sidrah Nausheen, FCPS

Academic Editor

PLOS ONE
---

## [Author Response · Author response to Decision Letter 1]

12 May 2023

Respected reviewers, 

Thank you for your feedback on the manuscript. This was helpful in shaping the manuscript. Based on the last feedback/comment pertaining to the alignment of objectives with the analysis, we have corrected this. We have provided a point-by-point response and a manuscript with tracked changes. 

Please let us know in the case of future queries. Thank you again for your time and feedback. 

Regards

Ameer Muhammad

---

## [Decision Letter · Decision Letter 2]

18 Jun 2023

PONE-D-22-17428R2Client-centered Counseling and Facilitation in Improving Modern Contraceptive Uptake in Urban slum of Karachi PakistanPLOS ONE

Dear Ameer mohammad 

Thank you for submitting your manuscript to PLOS ONE. After careful consideration, we feel that it has merit but does not fully meet PLOS ONE’s publication criteria as it currently stands. Therefore, we invite you to submit a revised version of the manuscript that addresses the points raised during the review process.

ACADEMIC EDITOR: the title on cover letter does not match with original article title. kindly correct it please explain client centered approach in detail. just listeneing and involving women in decision making does not work in our community where men has empowerment . what innovation are you bringing through this paper in previous literature most of the findings are same as already present in literarure . kindly highlight your innovation what new message are you giving to reader ? way forward ? Please submit your revised manuscript by 18th july 2023 . If you will need more time than this to complete your revisions, please reply to this message or contact the journal office at plosone@plos.org. Please include the following items when submitting your revised manuscript:A rebuttal letter that responds to each point raised by the academic editor and reviewer(s). You should upload this letter as a separate file labeled 'Response to Reviewers'.A marked-up copy of your manuscript that highlights changes made to the original version. You should upload this as a separate file labeled 'Revised Manuscript with Track Changes'.An unmarked version of your revised paper without tracked changes. You should upload this as a separate file labeled 'Manuscript'.If applicable, we recommend that you deposit your laboratory protocols in protocols.io to enhance the reproducibility of your results. Protocols.io assigns your protocol its own identifier (DOI) so that it can be cited independently in the future. For instructions see: https://journals.plos.org/plosone/s/submission-guidelines#loc-laboratory-protocols. Additionally, PLOS ONE offers an option for publishing peer-reviewed Lab Protocol articles, which describe protocols hosted on protocols.io. Read more information on sharing protocols at https://plos.org/protocols?utm_medium=editorial-email&utm_source=authorletters&utm_campaign=protocols.

We look forward to receiving your revised manuscript.

Kind regards,

Sidrah Nausheen, FCPS

Academic Editor

PLOS ONE

Journal Requirements:

Reviewers' comments:

Reviewer's Responses to Questions

**Comments to the Author**

1. If the authors have adequately addressed your comments raised in a previous round of review and you feel that this manuscript is now acceptable for publication, you may indicate that here to bypass the “Comments to the Author” section, enter your conflict of interest statement in the “Confidential to Editor” section, and submit your "Accept" recommendation.

Reviewer #1: All comments have been addressed

2. Is the manuscript technically sound, and do the data support the conclusions?

Reviewer #1: Yes

3. Has the statistical analysis been performed appropriately and rigorously? 

Reviewer #1: Yes

4. Have the authors made all data underlying the findings in their manuscript fully available?

Reviewer #1: Yes

5. Is the manuscript presented in an intelligible fashion and written in standard English?

Reviewer #1: Yes

6. Review Comments to the Author

Reviewer #1: (No Response)

7. PLOS authors have the option to publish the peer review history of their article (what does this mean?). If published, this will include your full peer review and any attached files.

Reviewer #1: No

---

## [Author Response · Author response to Decision Letter 2]

6 Jul 2023

Dear Editor,

Thanks for your feedback. We have attached the point-by-point response with the revise submission. 

Thank you again for valuable inputs which we strongly believe helped us in improving the reporting. 

Regards

Ameer Muhammad

---

## [Editor Report · Decision Letter 3]

7 Jul 2023

PONE-D-22-17428R3Client-centered Counseling and Facilitation in Improving Modern Contraceptive Uptake in Urban slum of Karachi PakistanPLOS ONE

Dear Dr. Muhammad,

Thank you for submitting your manuscript to PLOS ONE. After careful consideration, we feel that it has merit but does not fully meet PLOS ONE’s publication criteria as it currently stands. Therefore, we invite you to submit a revised version of the manuscript that addresses the points raised during the review process.

 1. conclusion should be concise relevant and should deliver key messages for the reader and future research. kindly write key messages in your conclusion and summarize it 

2. there is no reference no 46 cited in text . remove it from text as well. 

We look forward to receiving your revised manuscript.

Kind regards,

Sidrah Nausheen, FCPS

Academic Editor

PLOS ONE
---

## [Author Response · Author response to Decision Letter 3]

7 Jul 2023

Dear Editor, 

Thank you for your feedback. We have addressed both comments. The conclusion is now more concise, based on learning and limitations. Furthermore, reference 46 has also been added. 

Many thanks again. 

Regards

Ameer Muhammad

---

## [Editor Report · Decision Letter 4]

12 Jul 2023

Client-centered Counseling and Facilitation in Improving Modern Contraceptive Uptake in Urban slum of Karachi Pakistan

PONE-D-22-17428R4

Dear Ameer Muhammad,

We’re pleased to inform you that your manuscript has been judged scientifically suitable for publication and will be formally accepted for publication once it meets all outstanding technical requirements.

Kind regards,

Sidrah Nausheen, FCPS

Academic Editor

PLOS ONE
---

## [Editor Report · Acceptance letter]

19 Jul 2023

PONE-D-22-17428R4 

Client-centered Counseling and Facilitation in Improving Modern Contraceptive Uptake in Urban slum of Karachi Pakistan 

Dear Dr. Muhammad:

I'm pleased to inform you that your manuscript has been deemed suitable for publication in PLOS ONE. Congratulations! Your manuscript is now with our production department. 

Kind regards, 

on behalf of

Dr. Sidrah Nausheen 

Academic Editor

PLOS ONE